# Analysis of Electrical Resistivity Characteristics and Damage Evolution of Soil–Rock Mixture under Triaxial Shear

**DOI:** 10.3390/ma16103698

**Published:** 2023-05-12

**Authors:** Mingjie Zhao, Songlin Chen, Kui Wang, Gang Liu

**Affiliations:** 1Engineering Research Center of Diagnosis Technology and Instruments of Hydro-Construction, Chongqing Jiaotong University, Chongqing 400074, China; m.j.zhao@163.com (M.Z.); anhuiwk@163.com (K.W.); cqjtulg@163.com (G.L.); 2School of Civil Engineering and Architecture, Chongqing University of Science and Technology, Chongqing 401331, China

**Keywords:** soil–rock mixture (S-RM), electrical resistivity, damage model, triaxial shear, mechanical behavior

## Abstract

Construction of engineering structures in geomaterials with soil–rock mixture (S-RM) is often a challenging task for engineers. When analyzing the stability of the engineering structures, the mechanical properties of S-RM often receive the most attention. To study the mechanical damage evolution characteristics of S-RM under triaxial loading conditions, a modified triaxial apparatus was used to conduct shear test on S-RM, and the change of electrical resistivity was measured simultaneously. The stress–strain–electrical resistivity curve and stress–strain characteristics under different confining pressures were obtained and analyzed. Based on the electrical resistivity, a mechanical damage model was established and verified to analyze the damage evolution regularities of S-RM during shearing. The results show that the electrical resistivity of S-RM decreases with increasing axial strain and that the differences in decrease rates correspond to the different deformation stages of the samples. With the increase in loading confining pressure, the stress–strain curve characteristics change from a slight strain softening to a strong strain hardening. Additionally, an increase in rock content and confining pressure can enhance the bearing capacity of S-RM. Moreover, the derived damage evolution model based on electrical resistivity can accurately characterize the mechanical behavior of S-RM under triaxial shear. Based on the damage variable *D*, it is found that the damage evolution process of S-RM can be divided into a non-damage stage, a rapid damage stage and a stable damage stage. Furthermore, the structure enhancement factor, which is a model modification parameter for the effect of rock content difference, can accurately predict the stress–strain curves of S-RMs with different rock contents. This study sets the stage for an electrical-resistivity-based monitoring method for studying the evolution of internal damage in S-RM.

## 1. Introduction

Soil–rock mixture (S-RM) is widely distributed around the world to meet a wide range of architectural demands; “material origin” and “geological dynamic action formed by accumulation” jointly determine S-RMs in nature. Xu et al. [1] concluded that the origins of S-RM are very complex and mainly include the origin types of remnant slope accumulation, collapse accumulation, scour accumulation, glacial accumulation and artificial accumulation. Due to the complexity of geological origin and formation process, S-RM is generally different from traditional soil types. Xu et al. [2] and Afifipour et al. [3] distinguish S-RM from traditional geotechnical mass from the perspective of key physical indices and divide it into an emerging geotechnical medium system. Wang et al. [4] concluded that the main characteristics of S-RM include a complex composition of components, the mixing of different blocks with a range of shape and size, a stochastic distribution of rock blocks and an evident scale effect. In recent years, S-RM has been used in various applications such as earth–rockfill dams, high-fill works, subgrade works or foundation aggregates for construction work [5].

Among many factors affecting the mechanical properties of S-RM, the rock block content is always the first factor to be considered. Yao et al. [6] investigated the effect of rock content and shape distribution on the shear strength of S-RM under different stress conditions. The results showed that the contribution of contact force within gravel particles increased with increasing rock content. Before loading, the rock and soil in S-RM have a good coupling state; after being loaded, the breakage of coarse particles occurs, leading to the formation and expansion of microcracks. These anomalous changes are often regarded as the process of cumulative damage of S-RM and determine the magnitude of the load-bearing capacity of material and the destabilization damage process. In this paper, localized deterioration caused by the formation, convergence and propagation of microcracks inside the material is defined as damage. Therefore, accurate characterization of the damage evolution process of S-RM has gradually become a focus of interest for researchers.

Electrical resistivity is a physical quality of a material that characterizes the conductive properties of a material. For porous media such as soil and rock, their electrical resistivity is often considered as the representation of the spatial and temporal variability of many physical properties of soil (i.e., internal structure, water content or fluid composition) [7]. Electrical resistivity has been used by many researchers in various studies on soil [8,9], rock [10], coal [11,12] and concrete [13] to interpret the various properties of the research object. Khurshid and Afgan [14] investigated the effects of the injection of engineered water into carbonate reservoirs on electrical conductivity, ion mobility, electrical double layer thickness and the related oil recovery. Some quantitative and qualitative correlations between electrical resistivity and soil parameters such as cohesion, friction angle and plasticity index have been obtained. Moreover, the electrical resistivity characteristics of S-RM have also been initially investigated. According to the experimental results from Zhao et al. [15], variables such as rock content, compaction and water content all have an effect on the electrical resistivity variations of S-RMs. Moreover, electrical resistivity models, such as the series–parallel connection model [16], were established to characterize the relationship between the electrical resistivity and the physical property parameters of S-RM. Wang et al. [17] obtained the response regularity of the electrical resistivity of S-RMs with different rock contents and compactness during the water absorption–saturation process. Wang et al. [18] studied the variations in matrix suction and electrical resistivity in S-RM and found that there was a correlation between the matrix suction and the change in the electrical resistivity of S-RM.

Currently, most of the investigations conducted on the damage evolution of different materials are generally focused on brittle materials such as rock, concrete and coal. In a study on rock damage, Wang et al. [19] determined the damage variable of rock under loading conditions based on acoustic emission measurements, and the results showed that the damage variable increased gradually with the increasing deformation process. Zhang et al. [20] studied the variations in acoustic emission and infrared radiation of granite under uniaxial compression; they found that the complementary damage model based on sound–heat could accurately characterize the damage evolution characteristics of rock. Khurshid and Fujii [21] examined the influence of low-temperature CO_2_ on the decrease of formation breakdown pressure and the associated reservoir rock damage from a geomechanical prospective. Chung et al. [22] found that electrical resistivity had distinct responses to the damage state of concrete such as the freeze–thaw cycle and drying. Cao et al. [23] studied the variation in the electrical resistance of concrete under loading conditions and found that there was a synchronous relationship between the compressive damage and the change in concrete electrical resistance. When studying the relationship between concrete electrical resistivity and compressive damage, Zeng et al. [24] reported that the derived relationship between the damage variable and electrical resistivity is well correlated with the experimental data, in which electrical resistivity showed a trend of decreasing first and then increasing. Li et al. [25] applied acoustic emission to monitor the damage evolution process of coal mass induced by multi-stage loading and pointed out that the damage evolution in coal mass can be revealed and characterized by the acoustic emission. Similar conclusions had been drawn by Jia et al. [26] through a triaxial compression test, where the different damage evolution regularities of coal mass at different depths can be effectively comprehended by the acoustic emission. Based on the electrical resistivity testing data, the resistivity damage model of residual soil under uniaxial loading was proposed [27]; however, the general applicability of the damage model needs to be further explored. These investigations are capable of providing beneficial quantitative information for the damage assessment but are insufficient, as most of them rarely focus on the research of S-RM.

In summary, the electrical resistivity is closely related to the traditional geotechnical parameters of S-RM. Although many significant efforts have been made to investigate the electrical properties of S-RM, almost all the experiments mainly focus on the static measurement of indoor samples using the four-electrode or two-electrode methods. However, there is a lack of an in-depth study on the mechanical properties of S-RM based on the electrical resistivity method. Moreover, the quantitative study of the damage evolution of S-RM based on electrical resistivity in the process of shearing has not been reported, and the damage evolution characteristics of S-RM are the basis for understanding the formation and propagation of internal cracks. Therefore, it is of crucial significance to develop a method for characterizing the internal damage evolution of S-RM. 

This study aims to explore the influence of triaxial shear behavior on the electrical response to applied voltage and the damage evaluation method based on electrical resistivity monitoring. In this paper, a modified triaxial apparatus was used to measure the electrical resistivity of S-RM subjected to triaxial shear, and the electrical resistivity method was used to evaluate the change in damage variable *D*, which quantifies the degree of damage. The relationship between the electrical resistivity and the stress–strain of S-RM was derived. A mechanical damage model based on the electrical resistivity of S-RM was established, and the calculated results of the model were compared with the measured results to verify the reliability of the damage model. Additionally, the damage evolution regularity of S-RM was discussed from the perspective of electrical resistivity, which enriches the application of the electrical resistivity method in the study of S-RM damage. Based on the electrical resistivity results, we modified the established damage model to meet the mechanical damage analysis of S-RM with different rock contents.

## 2. Materials and Methods

### 2.1. Experimental Setup

In order to measure the electrical resistivity of S-RM samples under triaxial shear, the conventional triaxial apparatus was modified. The schematic drawing and experimental setup of the modified triaxial apparatus are illustrated in Figure 1. The apparatus can measure the change in the electrical resistivity of the sample during shearing simultaneously. The modified triaxial apparatus was manufactured by the Yongchang Science and Education Instrument Manufacturing Co., Ltd., Liyang, Jiangsu, China. It consists of four components: axial loading equipment, pressure chamber, confining pressure and in–out water control system and data acquisition system. The inner dimensions of the cylindrical sample are 100 mm (diameter) × 200 mm (height). The maximum confining pressure that the pressure chamber can bear and the maximum axial force that the axial loading system can apply are 2000 kPa and 250 kN, respectively. Various test variables such as confining pressure, axial force, electrical resistivity, displacement, pore pressure, etc., are recorded and stored by the data acquisition control system; the confining pressure and axial loading can be controlled automatically according to their target values. To measure the electrical resistivity, two metal electrodes are attached to the top cap and base pedestal of the triaxial apparatus. The electrodes are made of porous stainless-steel discs with a diameter of 100 mm. Two thin copper wires wrapped with insulating material lead out from the base pedestal of the triaxial apparatus. The copper wires in the pressure chamber are connected to the upper and lower electrodes, and the copper wires outside the pressure chamber are connected to the electrical resistivity measurement device. 

To date, the two-electrode and four-electrode methods have been adopted by most researchers for the laboratory electrical resistivity measurement. In the four-electrode method the most important consideration is related to the disturbance of test sample introduced by the insertion of electrical probes. Moreover, the four-electrode method is difficult to adopt in triaxial tests due to the difficulty in determining the distance between the electrodes [28]. In this study, the two-electrode method is adopted for the electrical resistivity measurement. For the two-electrode method, since the same pair of electrodes are used for both the injection of current and the measurement of potential difference, a polarization effect usually occurs. In the present work, the polarization effect can be avoided by changing the polarity of the current electrode. In order to reduce the contact resistance between the two ends of the sample and the metal electrodes, both ends of the sample were evenly coated with conductive graphite before loading. The power supply provides an output voltage of 9 V.

### 2.2. Materials

The formation lithology of the Chongqing region, China, consists mainly of silty clay, sandstone, mudstone and sandy mudstone. These materials are widely used to construct levees and earthen dams in China and are internally unstable in essence. In order to facilitate the local selection of the experimental materials, a mixture of mudstone and clay from this region was selected as the raw material for this study. The raw materials retrieved from the site were sieved and then secondly proportioned. The particles with a diameter of 0–20 mm were then selected as the experimental materials (see Figure 2a). The soil–rock threshold (S-RT) can be used to distinguish the particle size limits of “soil matrix” and “rock block” in S-RM; it is a very significant physical property index of S-RM. According to some previous studies [29,30], the method of determining the S-RT has been given. Medley proposed that the S-RT is not a fixed value but should be a variable value related to the area of the S-RM, denoted as 0.05$L_{c}$, where $L_{c}$ is the scale of engineering features such as the tunnel diameter, slope height and laboratory sample diameter. Taking into account the geomechanical contribution of the blocks, Zhang et al. [30] and Tu et al. [31] gave the S-RT criterion as 5 mm. Meanwhile, in this paper, considering the size of the laboratory test samples, the S-RT criterion of 5 mm is adopted. This means that the S-RM could be defined as “soil” if the particle size less than 5 mm, whereas it could be defined as “rock” if the particle size greater than 5 mm. The main physical property indexes of S-RM retrieved from the site are summarized in Table 1. Based on X-ray diffraction, the specific mineral composition of the rock is obtained, as shown in Table 2. Generally, the mechanical properties of S-RM depend on the joint action of soil and rock only when the rock content is between 25% and 75%. Considering the rock content of the raw material, four rock contents of 20%, 30%, 40% and 50% are set, and the corresponding particle size distribution of each rock content is shown in Figure 2b. Figure 3 shows an SEM image of the soil particles’ surfaces. We found that the surfaces of the particles had an irregular scaly stacking structure with different sizes, and a small number of holes presented after local amplification.

In addition, the maximum dry density at four rock contents was determined through the proctor compaction test. The target dry density is determined by considering the maximum dry density of raw materials and S-RM with different rock contents, that is, the target dry density is 1.95 g/cm^3^ with 50% rock content and the target dry density of remaining rock content decreases in order. The water content of the sample preparation is 5%. The values of the main physical parameters of the S-RM samples with different rock contents are shown in Table 3.

### 2.3. Test Procedure

Triaxial compression tests were conducted under the conventional path, which consists of three basic steps. The first step is the preparation of the S-RM samples. Firstly, soil and rock are oven-dried at 103 °C for 24 h; the amounts of soil and rock required to achieve the target dry density are weighed according to the particle size distribution curve with different rock contents. Secondly, the required amount of water (gravimetric water content of 5%) is weighed and mixed evenly with the S-RM. Finally, the prepared S-RM is put into a cylindrical mold (which has a height of 200 mm and a diameter of 100 mm) in three layers and compacted in a stratified manner. In order to enhance the interlayer connectivity of the samples, artificial chiseling is applied to the interlayer contact surfaces. Double rubber membranes are utilized to prevent the membrane from being punctured by the sharp corners of irregular-shaped rock blocks.

The second step was to saturate and consolidate the S-RM samples. The prepared S-RM samples were installed on the base pedestal of the triaxial apparatus. In the whole process of installation, the disturbance to the sample was minimized. After the installation, in order to speed up the saturation process, the backpressure saturation method was adopted to saturate the S-RM samples. After the S-RM samples were saturated, the confining pressure was adjusted to the design value and then consolidation tests were conducted. During consolidation, pore water was allowed to be discharged and the confining pressure remained stable. When the drainage volume changes very little and almost no longer changes, the consolidation of the S-RM samples was considered to be completed. In order to capture the electrical resistivity data of tested sample more accurately, the consolidation tests started when the data were stable.

The third step was to conduct a shear test on the S-RM samples. Given the long time required for the shearing test, the time interval for the electrical resistivity data acquisition was set to 120 s. After the consolidation, the sample underwent a slight deformation, resulting in a decrease in sample height. Therefore, it was necessary to make full contact between the pressure shaft and the top cap before starting the shearing test. Once the electrical resistivity data remained stable, drained triaxial shear tests were conducted on the S-RM samples under different confining pressure levels (250 kPa, 400 kPa, 550 kPa and 700 kPa). The shear mode is strain control, and the shear rate is set to 0.13 mm/min. When the axial strain reached 15% (axial displacement of 30 mm), the shearing process was terminated. As a result, the deviatoric stress, axial strain and electrical resistivity were recorded and stored by the data acquisition system in the shearing process. In addition, we noted that the contact resistance in this experiment was diminished since the polishing treatment of the electrode surfaces was conducted after each measurement. Figure 4 shows all the S-RM samples after the triaxial shear was completed.

## 3. Experimental Results and Discussion

### 3.1. Stress–Strain–Electrical Resistivity Curve

Through the triaxial test results, it is found that the stress–strain–resistivity variation regularities of the S-RM samples with the same rock contents under four confining pressures are similar. Therefore, some typical resistivity variation curves are selected for response regularity analysis. Figure 5 plots the typical response of electrical resistivity with increasing deviator stress. It can be observed that the electrical resistivity varies continuously with increasing deviator stress, which also provides the possibility for the establishment of a subsequent electrical resistivity damage model.

In general, in the process of complete strain, the electrical resistivity curve of the S-RM samples shows a gradually decreasing trend that presents an irregular “slightly concave shaped”. The volumetric strain curves of the loaded coal and rock mass have obvious inflection points; based on this, for convenience, their stress–strain curves are commonly divided into different deformation stages in existing experiments to study the electrical resistivity variation properties [10,11,12]. In this paper, there is no obvious inflection point in the volumetric strain curve of S-RM, as shown in Figure 6. Therefore, combining the electrical resistivity evolution characteristics of S-RM samples and the classification methods of different deformation stages of coal and rock mass, the deformation process of samples is roughly divided into three distinct deformation stages, and there are differences in the reduction rate of electrical resistivity under each stage. In the initial compaction stage (I), there is a short and rapid decrease in electrical resistivity; in the elastic deformation stage (II), the reduction rate of the electrical resistivity value diminishes with increasing axial strain; in the yield deformation stage (III), the reduction rate further decreases. The electrical resistivity of S-RM samples generally follows the following variation regularity: it initially decreases rapidly, then declines slowly and finally reduces gently. As reported in some triaxial shear test studies [28,32], the obtained stress–strain curves are also divided into several distinct deformation stages.

However, the electrical resistivity response mechanism of S-RM is distinct in different deformation stages. It can be surmised that the decrease in electrical resistivity under the initial compaction stage is dependent upon the closure of pores and microcracks in the sample and the discharge of high-resistance gas. The deceleration of electrical resistivity under the elastic deformation stage is mainly influenced by the increase in sample saturation and the electron conductivity. The slow decrease in electrical resistivity during the yield stage can mainly be attributed to the progressive discharge of pore water in the S-RM sample and the qualitative change in its internal structure. There are similar assumptions to explain the change in soil electrical resistivity under freeze–thaw cycles in the field of soil damage evaluation [33].

### 3.2. Stress–Strain Characteristics of S-RM

The stress–strain curves of each rock content under four confining pressures are plotted in Figure 7. By analyzing and summarizing the stress–strain curves, three different variation regularities can be derived, as shown in Figure 8a. Such variation characteristics are consistent with the results obtained by Zhang et al. [34]. This implies that the stress–strain characteristics of S-RM gradually change from a slight strain softening to a strong strain hardening as the confining pressure increases. The increasing rate of the deviator stress is fast at the beginning and then slows down soon after. Unlike rock mass, the stress–strain curve of S-RM has no obvious peak stress phenomenon. This indicates that S-RM has the strength property of being able to withstand a certain loading even in the late stage of strain. Therefore, it can be predicted that if the confining pressure is further increased, then the strain hardening characteristics of S-RM will be further enhanced. All of the S-RM samples show radial dilatation deformation that is more prominent in the middle of the samples, which can be attributed to the bottom layer of the samples withstanding the major compaction energy during the sample preparation and the pressure shaft acting directly on the top of the samples. Furthermore, in the late stage of strain, the energy dissipation caused by the partial coarse particle breakage is compensated for by adjusting the internal structure. Therefore, the apparent particle breakage does not result in a corresponding reduction in the deviator stress.

Figure 8b shows the maximum deviator stress that S-RM samples can withstand under different confining pressures. When the confining pressure is constant, the maximum deviator stress increases with the increase in rock content. We refer to the difference in shear strength as the “enhancement effect of rock block”. The cause of such a phenomenon is that the skeleton structure of the rock blocks has been generated inside the S-RM samples and has the capacity to resist deformation under external loading. Although the samples with different rock contents have different initial compactions, it is undeniable that increasing rock content, skeleton structure formation and the interlock effect among the coarse particles play a significant role in enhancing the strength characteristics of S-RM. Moreover, with the increase in confining pressure, the corresponding deviator stress of the samples with the same rock content also presents an incremental trend.

### 3.3. Establishment of Mechanical Damage Model

On the basis of the original electrical resistivity data, we further establish the damage model of S-RM. As mentioned earlier, the variation characteristics for electrical resistivity and deviator stress with axial strain provide the possibility for the establishment of a damage model. Specifically, if the sample is in the initial compaction stage, there is almost no structure damage inside it and the elastic deformation can be restored to the initial state when no external loading is applied. In this case, we suppose that no damage occurs inside the sample during the compaction and elastic deformation stages. From the test results (see Figure 5 for part), it can be seen that the yield deformation for the studied sample starts to occur when the axial strain is about 2%. Therefore, the axial strain of 2% is taken as the initial strain for damage propagation, which is denoted as $\varepsilon_{2}$; the corresponding electrical resistivity is denoted as $\rho_{\mathrm{ini}}$.

For the convenience of analysis, it is assumed that the damaged sample consists of the middle section with structural damage and the two ends with structural integrity [27]. The structural damage mainly includes the changes in the failure morphology, deformation behavior and other characteristics, such as shear dilatation and strain hardening, which are closely related to the evolution of the meso-structure of S-RM. Assuming that the cross-sectional area and the height of the S-RM sample in the initial stage of damage propagation are *S*_0_ and *L*_0_, respectively, the heights of two ends with intact structure are $L_{0}k_{1}$ and $L_{0}k_{2}$, where $k_{1}$ and $k_{2}$ are the proportion coefficients of the height of the two ends to the height of the entire sample, respectively. The height of the intermediate damage section is $L_{0}\beta$, where β is the proportion coefficient of the middle section to the height of the entire sample. The damage mode of the S-RM sample is shown in Figure 9. $k_{1}$, $k_{2}$ and β satisfy the following formula:(1)$$k_{1}+k_{2}+\beta =1$$

Assuming that the S-RM sample is a uniform conductive cylinder when the strain is 2%, then the resistance can be written as follows:(2)$$R=\frac{\rho L_{0}}{S_{0}}=\frac{\rho L_{0}^{2}}{V_{0}}$$
where R is the resistance of the S-RM sample, Ω; ρ is the measured electrical resistivity, Ω·m; $V_{0}$ is the volume of the S-RM sample in the initial stage of damage propagation, $m^{3}$.

The sample height will gradually change during the shearing process, so the measured electrical resistivity may be different from the actual resistivity. Considering the effect of the dimensional change, the amended electrical resistivity value is obtained as:(3)$$\rho_{r}=\frac{\rho }{{\left(1-\varepsilon \right)}^{2}}$$
where $\rho_{r}$ is the realistic electrical resistivity, Ω·m; ε is the axial strain, %.

Equation (2) can be written as follows:(4)$$R=\frac{\rho_{r}{\left(1-\varepsilon_{2}\right)}^{2}L_{0}^{2}}{V_{0}}$$

When the axial strain of the S-RM sample is ε, the resistance can be written as follows:(5)$$R_{\phi }=k_{1}\frac{\rho_{r,\mathrm{ini}}L_{0}^{2}{\left(1-\varepsilon \right)}^{2}}{V_{0}}+k_{2}\frac{\rho_{r,\mathrm{ini}}L_{0}^{2}{\left(1-\varepsilon \right)}^{2}}{V_{0}}+\beta \frac{\rho_{r,\mathrm{ini}}L_{0}^{2}{\left(1-\varepsilon \right)}^{2}}{\phi_{v}V_{0}}=\frac{\rho_{\phi }L_{0}^{2}}{V_{0}}$$
where ϕ is the extent of damage development; $\rho_{\phi }$ is the damage electrical resistivity (measured value); $\rho_{r,\mathrm{ini}}$ is the realistic electrical resistivity under initial damage, which is expressed as $\rho_{r,\mathrm{ini}}=\frac{\rho_{\mathrm{ini}}}{{\left(1-\varepsilon_{2}\right)}^{2}}$; $\phi_{v}$ is the damage volume coefficient of the sample.

Equation (5) can be further simplified:(6)$$\frac{\rho_{\phi }{\left(1-\varepsilon_{2}\right)}^{2}}{\rho_{\mathrm{ini}}{\left(1-\varepsilon \right)}^{2}}=\beta (\frac{1}{\phi_{v}}-1)+1$$

The damage volume coefficient $\phi_{v}$ can be further expressed as:(7)$$\phi_{v}=\frac{V_{a}}{V_{0}}=\frac{L_{0}(1-\varepsilon )S_{a}}{L_{0}S_{0}}=(1-\varepsilon )\phi_{s}$$
where $V_{a}$ is the volume of the S-RM sample; $S_{a}$ is the average cross-sectional area of the S-RM sample; $\phi_{s}$ is the damage coefficient. Substituting Equation (7) into Equation (6):(8)$$\frac{\rho_{\phi }{\left(1-\varepsilon_{2}\right)}^{2}}{\rho_{\mathrm{ini}}{\left(1-\varepsilon \right)}^{2}}=\beta \left(\frac{1}{(1-\varepsilon )\phi_{s}}-1\right)+1$$

The effective conductivity coefficient $\phi_{\rho }$ is introduced, and its expression is:(9)$$\phi_{\rho }=\frac{\rho_{\phi }{\left(1-\varepsilon_{2}\right)}^{2}}{\rho_{\mathrm{ini}}{\left(1-\varepsilon \right)}^{2}}$$

Therefore, the damage variable *D* can be derived from the damage coefficient $\phi_{s}$, which can be written as follows:(10)$$D=1-\phi_{s}=1-\frac{1}{\left(1-\varepsilon )(1+\frac{\phi_{\rho }-1}{\beta }\right)}$$

Neglecting the geometric effects due to deformation, Equation (10) can be simplified:(11)$$D=1-\phi_{s}=1-\frac{1}{\left(1+\frac{\phi_{\rho }-1}{\beta }\right)}$$

According to the effective stress principle, the axial strain of the damaged material under stress σ is equal to that of undamaged material under effective stress $\sigma^{\prime }$. We obtain
(12)$$\varepsilon =\frac{\sigma }{E^{\prime }}=\frac{\sigma^{\prime }}{E}$$
where E and $E^{\prime }$ are the elastic modulus of the S-RM sample in the undamaged and damaged states, respectively.

The effective stress $\sigma^{\prime }$ and the damage variable *D* satisfy the following expression:(13)$$\sigma^{\prime }=\frac{\sigma }{1-D}$$

Substituting Equation (12) into Equation (13):(14)σ=(1−D)Eε

Substituting Equation (11) into Equation (14), the mechanical damage model of S-RM based on electrical resistivity can be written as follows:(15)$$\sigma =\frac{1}{\left(\frac{\phi_{\rho }-1}{\beta }+1\right)}E\varepsilon$$

The coefficient β represents the proportional magnitude of the damaged section in the entire sample, which changes gradually with increasing axial strain. Previous research had shown that the total crack area in S-RM samples under triaxial deformation gradually tends to a certain value with the increase in strain, which follows the following variation regularity: it initially increases rapidly and then increases gently; such changes in microcracks always occur in the range of 6~8% axial strain [35]. Therefore, in this paper, it is assumed that the coefficient β hardly varies in the late stage of axial strain and is a fixed value. The critical value of the abovementioned axial strain is taken as 6%, which is denoted as $\varepsilon_{6}$.

When the axial strain $\varepsilon <\varepsilon_{2}$, β = 0; when the axial strain ε is in the range of $\varepsilon_{2}\sim \varepsilon_{6}$, β increases monotonically from 0 to the maximum value $\beta_{\max}$; when the axial strain $\varepsilon >\varepsilon_{6}$, β = $\beta_{\max}$. Therefore, a piecewise function is used to describe the variation characteristics of coefficient β:(16)$$\beta (\varepsilon )=\left\{ \begin{array}{ll} 0 & \;(0\leq \varepsilon \leq \varepsilon_{2})\\ \beta_{\max}\sqrt{\frac{\varepsilon -\varepsilon_{2}}{\varepsilon_{6}-\varepsilon_{2}}} & \;(\varepsilon_{2}<\varepsilon \leq \varepsilon_{6})\\ \beta_{\max} & \;(\varepsilon >\varepsilon_{6}) \end{array}\right.$$

The maximum value $\beta_{\max}$ can be obtained as follows:(17)$$\beta_{\max}=\phi_{\rho ,\max}-1$$
where $\phi_{\rho ,\max}$ can be derived from Equation (9) by taking the maximum value of ε, $\varepsilon_{15}$.

In order to verify the reliability of the established damage model, the stress–strain curves of S-RM samples with 20% rock content under four confining pressures obtained from the test are compared with the calculated results from the damage model proposed in this paper, and the results are shown in Figure 10.

As expected, the stress–strain results calculated by the electrical resistivity damage model are in good agreement with the experimental results. Only when the confining pressure is 400 kPa is there a large difference between the above two results under large axial strain. The difference between the calculated value and the measured result is perhaps due to the sharp increase in deviator stress at 400 kPa and the limitation of the measurement accuracy caused by the test. This indicates that the reliability of the damage model still needs to be further refined due to the presence of partial assumptions during the model derivation. On balance, we believe that the damage model based on electrical resistivity obtained in this study can not only better characterize the internal structure damage of S-RM but also further validate the general applicability of the damage model proposed in the literature [24].

### 3.4. Damage Evolution Analysis of S-RM

Electrical resistivity is an indicator of the generation of structural damage inside the geomaterials until the occurrence of macroscopic damage by monitoring the closure of pores and the generation and propagation of microcracks, which can effectively evaluate the damage degree of geomaterials. Moreover, this experiment seems to prove the conclusion in most of the literature that electrical resistivity can accurately characterize the mechanical behavior of soil. Through the variation curve of damage variable *D*, the damage evolution characteristics of S-RM are analyzed from the perspective of the changes in internal microstructure. Figure 11 plots the variation curve of damage variable *D* during triaxial shear (confining pressure 700 kPa) for S-RM samples with 20% rock content. The maximum value of *D* is about 0.483. The stress–strain curve and damage variable–strain curve in Figure 11 are obtained from the experimental data in this paper and Equation (11), respectively.

As shown in Figure 11, the damage evolution of S-RM under triaxial shear can be divided into three stages: the non-damage stage, the rapid damage stage and the stable damage stage. As deformation grows, the damage of the S-RM sample also means that it is gradually compressed. During the non-damage stage, it can be seen that the damage variable *D* is 0 and that the damage–strain curve varies horizontally, which may be due to the fact that there is no structural damage such as particle breakage and microcrack formation in the sample when the sample is in the initial compaction and elastic deformation stages.

In the stage of rapid damage, the sample has entered the yield deformation stage, the damage variable *D* exhibits a trend of rapid increase and the damage–strain curve is a convex arc. It can be seen that the increasing rate of damage variable D decreases gradually. The interparticle contact forces increase further in the yield deformation stage, and coarse particles with their own defects in the S-RM start to break due to the stress concentration, so the damage can occur rapidly. With the increase in axial strain, the greater inter-particle contact force makes the existing defects of the particles further develop or causes new defects and the particle breakage rate increases, accompanied by the gradual propagation of microcracks. Compared with coarse particles, the fine particles formed after particle breakage have a lower breakage potential and are more difficult to break.

In the stage of stable damage, the damage variable *D* exhibits a steadily increasing trend and the gradient of the damage–strain curve tends to be stable. Because the coarse and fine particles have been broken to different degrees, a more uniform inter-particle contact force is formed, which leads to a decreasing possibility for massive breakage. Meanwhile, the internal microcracks of S-RM gradually increase until the formation of shear localized bands. Although there is still the particle breakage phenomenon at this stage, it seems that the propagation of the crack has a more significant effect on the damage. Under the combined effect of particle breakage and microcracks, the corresponding damage variable *D* presents an incremental trend.

Compared with that in the S-RM with a confining pressure of 700 kPa, the maximum damage variable of S-RM with lower confining pressure changes, which exhibits a slight increase with the decrease in confining pressure. The maximum damage variables of the sample with 20% rock content under the confining pressures of 250 kPa, 400 kPa and 550 kPa are 0.512, 0.498 and 0.490, respectively. For the S-RM, the confining pressure level can not only enhance the compactness of the sample but also may improve the overall strength effectively, so the degree of damage varies. Due to the difference in the continuity of the material composition between rock and S-RM, the damage evolution regularity of rock is significantly distinct from that of S-RM. The damage of rock under load was investigated in the literature [18]. They found that there is a turning point in the curve for the damage variable as a function of strain. There is no significant change in the damage variable before the turning point, but above this point the damage variable rapidly rises to the critical value.

### 3.5. Modification of Effect of Rock Content Difference

The results of previous studies indicate that the existence of rock blocks not only results in the enhancement of the compressive strength but also affects the difference in the electrical resistivity for S-RM samples [15,31]. Table 4 summarizes the statistics of the electrical resistivity values measured after the saturation of S-RM samples with four rock contents. It can be found that the range (difference between the maximum and minimum), standard deviation and coefficient of variation of each rock content are small, suggesting that all electrical resistivity values are relatively stable and have no significant fluctuations. Thus, the mean is chosen as the electrical resistivity value for each rock content after saturation, which is the comprehensive performance of both rock content and compaction.

According to the electrical resistivity values in Table 4, the structure enhancement factor of S-RM is introduced, which is used as a model modification parameter to characterize the effect of the rock content difference, denoted as *S*(*R*), and its expression is:(18)$$S(R)=\frac{ER(R)}{ER(20)}$$
where ER(20) is the electrical resistivity of S-RM with rock content 20% after saturation; ER(R) is the electrical resistivity of S-RM with different rock contents of 30%, 40% and 50% after saturation.

Based on the derived structure enhancement factor *S*(*R*) and the established damage model, the stress–strain curves of S-RM with different rock contents (30%, 40% and 50%) can be obtained. One confining pressure is selected for analysis for each rock content, as shown in Figure 12.

From Figure 12, it can be seen that the calculated values for the damage model after the introduction of the modification parameter have a good correlation with the experimental data for S-RM. Through simultaneous measurement of electrical resistivity in the triaxial shear test, the stress–strain curves of S-RM with different rock contents can be accurately predicted. Therefore, it can be considered that the electrical resistivity signals can effectively characterize and evaluate the internal damage degree of S-RM.

It should be noted that the coefficient β, which characterizes the magnitude of the middle-damaged section of the S-RM sample, is a piecewise function that has been simplified in the calculation process. In the actual shear process, the damage of the S-RM sample exists all the time, which increases gradually from early to late shear, and is far more complex than the monotonic increase assumed in this work. To well reveal the role of rock block in affecting damage behavior, the damage mechanism of the S-RM samples with various rock contents needs to be further discussed.

In summary, the experimental results show that the electrical resistivity is highly correlated with internal damage. Both the existence and the development of structural damage can be reflected by the electrical resistivity measurement of the S-RM models, which provides implications for developing an electrical-resistivity-based method for monitoring the structural failure in earth–rockfill dams. Therefore, compared with acoustic emission, infrared radiation and ultrasonic measurement, the electrical resistivity technique is more suitable for in situ testing in the field and has prospects for application in the study of geotechnical mechanics.

## 4. Conclusions

In order to explore the feasibility of the electrical resistivity method to characterize the damage behavior of S-RM under triaxial shear, the relationship between the change in electrical resistivity and the damage evolution of S-RM was studied. Under the applied voltage, the electrical resistivity response of S-RM samples was collected during triaxial deformation. The electrical resistivity variation regularities and stress–strain characteristics of S-RM under stress were explored using modified triaxial apparatus. The mechanical damage model, the damage evolution process and the modification of the effect of rock content difference were investigated and discussed. The main conclusions are as follows:(1)During the triaxial loading process the electrical resistivity of S-RM varies in stages with the axial strain, which follows the following variation regularity: it initially decreases rapidly, then declines slowly and finally reduces gently. The response mechanisms of electrical resistivity under different deformation stages are distinct.(2)The stress–strain characteristics of S-RM gradually change from a slight strain softening to a strong strain hardening as the confining pressure increases. Under the same confining pressure, the deviator stress that S-RM samples can withstand is not the same; the more the rock content, the higher deviator stress is. With the increase in confining pressure, the corresponding deviator stress of the samples with the same rock contents increases gradually.(3)The electrical-resistivity-based mechanical damage model for S-RM can accurately characterize the degree of structure damage. This mathematical model agrees with the experimental results. The damage evolution of S-RM under triaxial shear can be divided into three stages: a non-damage stage, a rapid damage stage and a stable damage stage. In addition, the modified model can accurately predict the mechanical behavior of S-RM with different rock contents. These discoveries can provide a basis for further research on the application of electrical resistivity.

## Figures and Tables

**Figure 1 materials-16-03698-f001:**
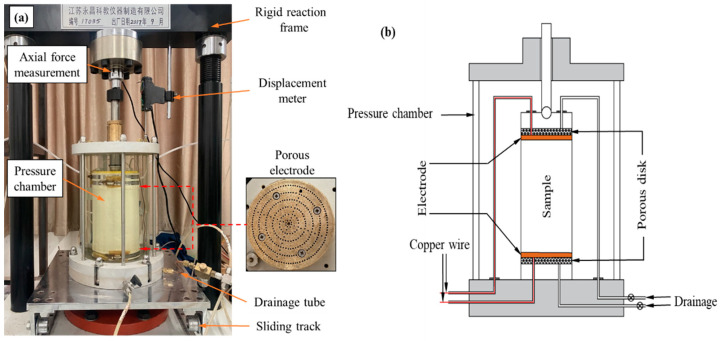
(**a**) Experimental setup. (**b**) Schematic drawing of the modified triaxial apparatus.

**Figure 2 materials-16-03698-f002:**
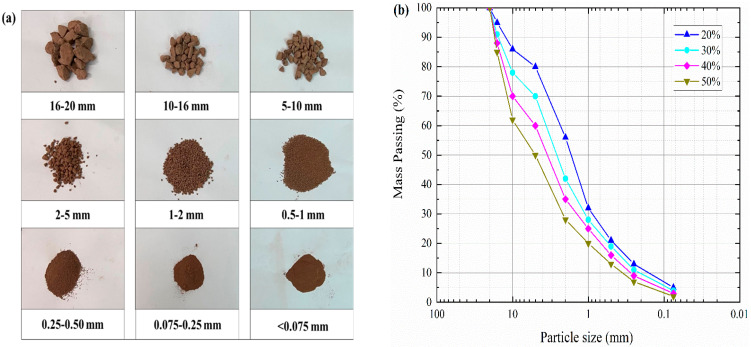
Material used for S-RM samples. (**a**) Photographs of each particle group. (**b**) Particle size distribution.

**Figure 3 materials-16-03698-f003:**
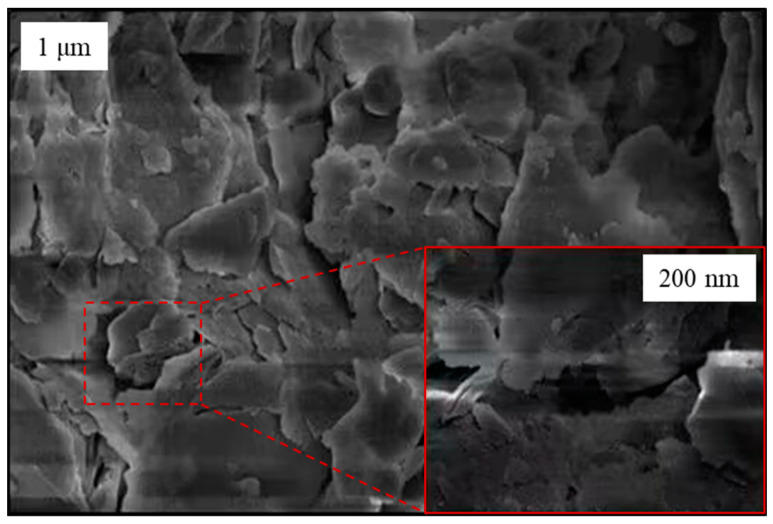
SEM image of particles’ surfaces.

**Figure 4 materials-16-03698-f004:**
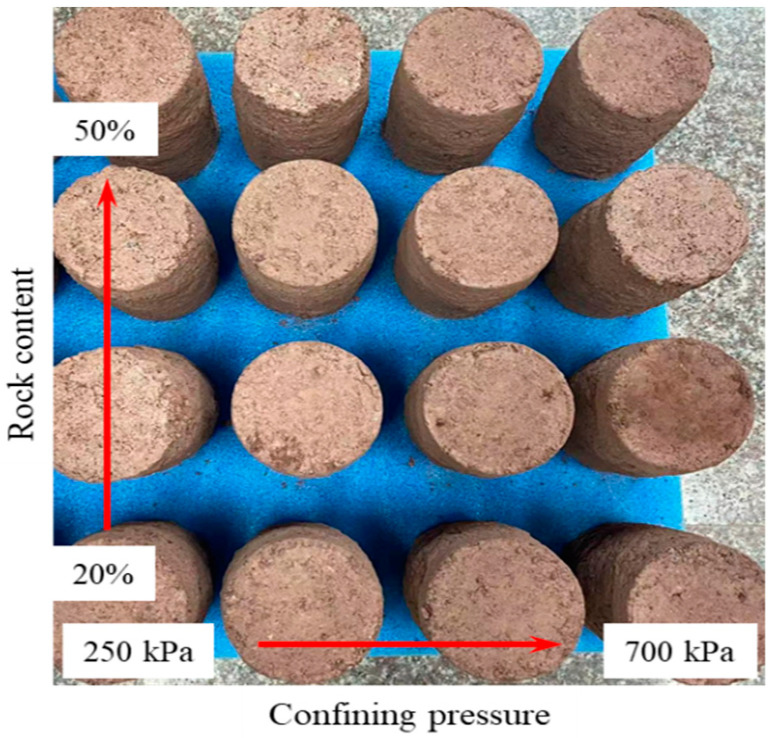
Photograph of S-RM samples after the shear.

**Figure 5 materials-16-03698-f005:**
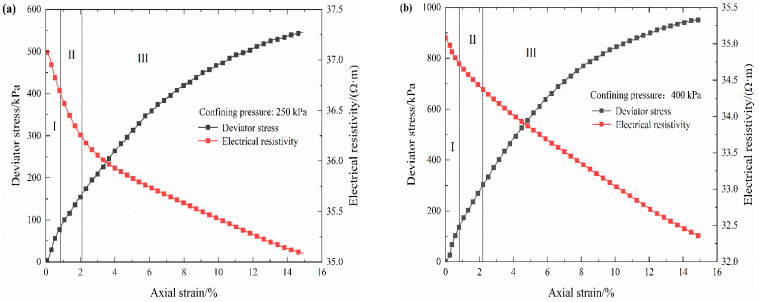
Stress–strain–electrical resistivity curves of S-RM samples. (**a**) Rock content 20%. (**b**) Rock content 30%. (**c**) Rock content 40%. (**d**) Rock content 50%.

**Figure 6 materials-16-03698-f006:**
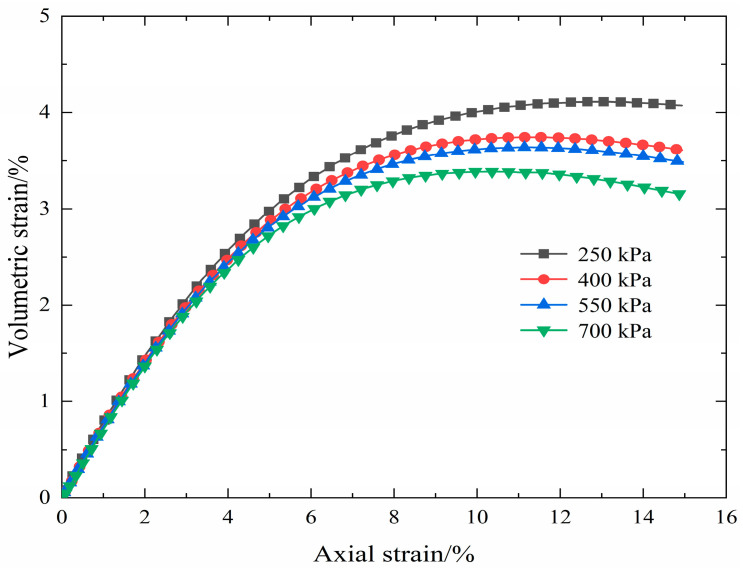
Volumetric strain curves of S-RM (rock content 40%).

**Figure 7 materials-16-03698-f007:**
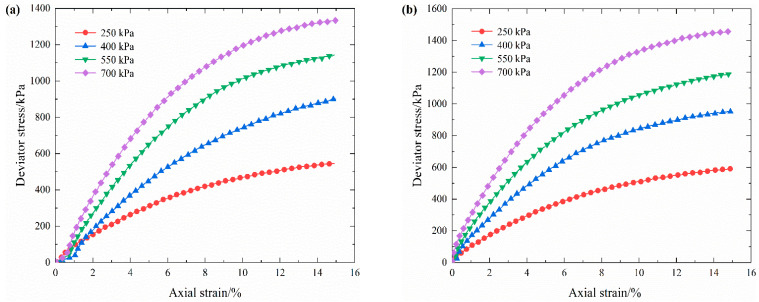
Stress–strain curves of S-RM for different confining pressures. (**a**) Rock content 20%. (**b**) Rock content 30%. (**c**) Rock content 40%. (**d**) Rock content 50%. The red circle symbols, the blue triangle symbols, the green inverted triangle symbols and the purple rhombus symbols represent confining pressures of 250 kPa, 400 kPa, 550 kPa and 700 kPa, respectively.

**Figure 8 materials-16-03698-f008:**
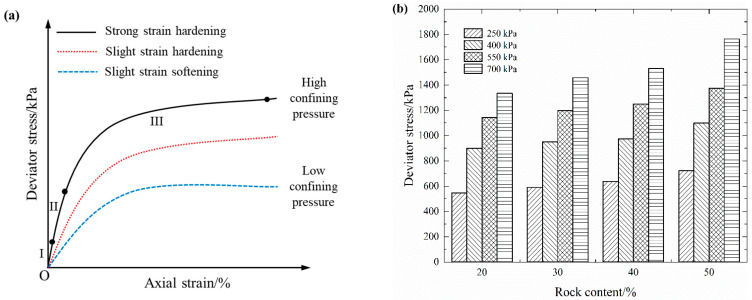
Variation regularities of deviator stress. (**a**) Variation of deviator stress with axial strain. (**b**) Variation of deviator stress with rock content.

**Figure 9 materials-16-03698-f009:**
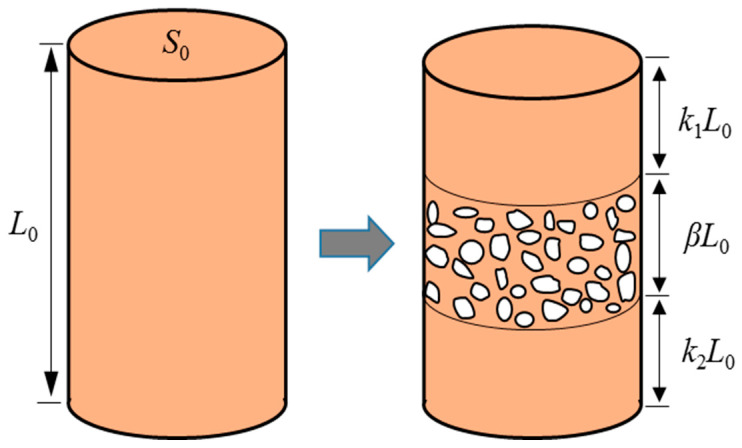
The schematic diagram of the S-RM damage mode.

**Figure 10 materials-16-03698-f010:**
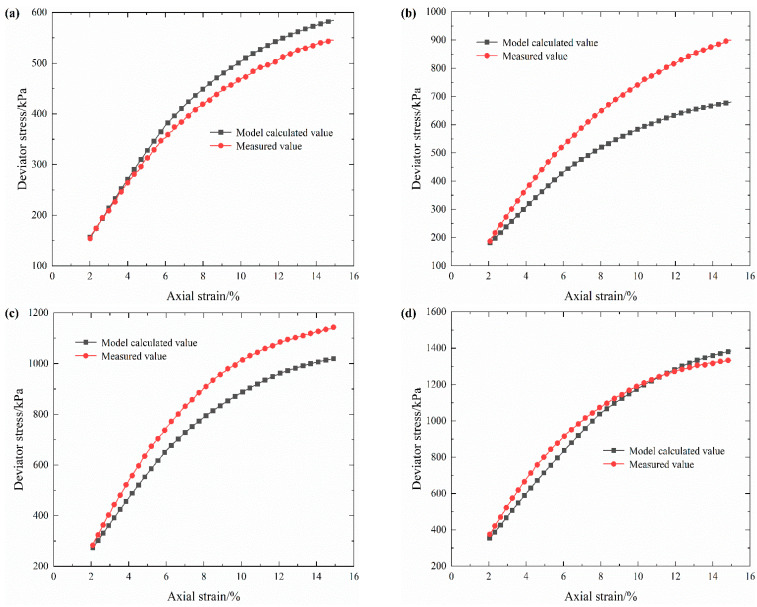
Measured results and calculated values from the damage model under different confining pressures. (**a**) 250 kPa. (**b**) 400 kPa. (**c**) 550 kPa. (**d**) 700 kPa.

**Figure 11 materials-16-03698-f011:**
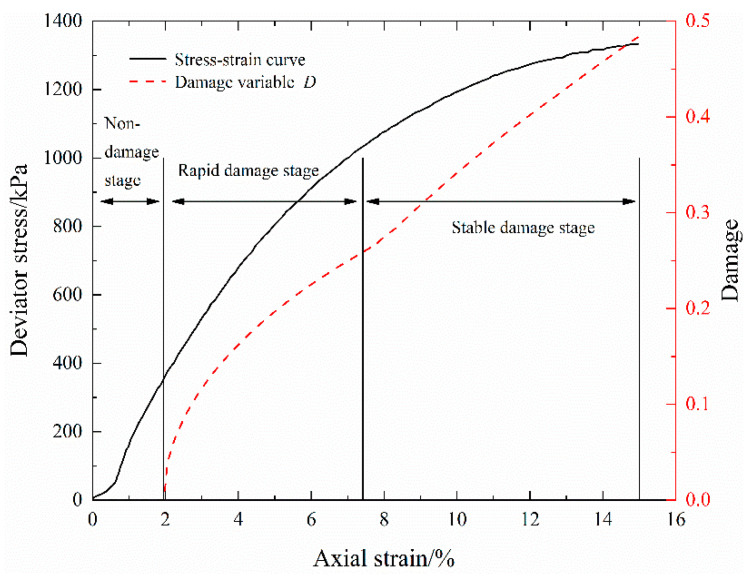
Variation curve of damage variable *D* during triaxial shear.

**Figure 12 materials-16-03698-f012:**
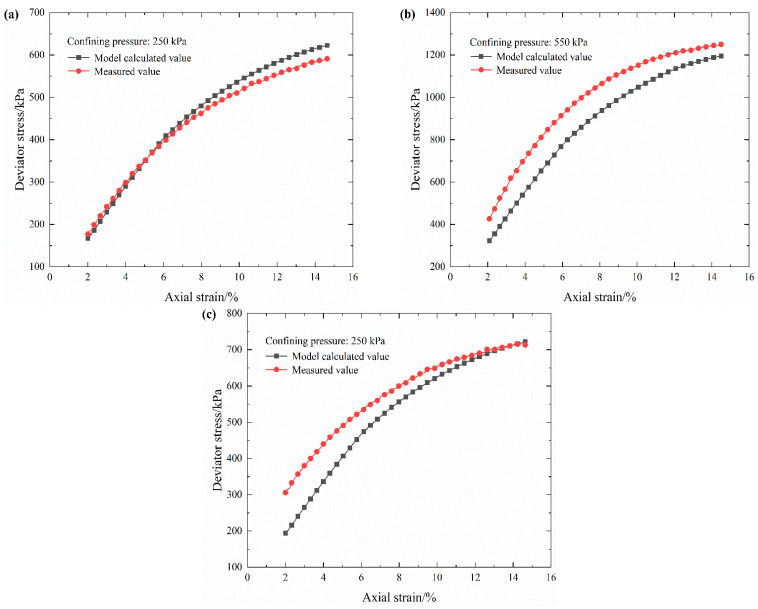
Measured results and calculated values of the damage model under partial confining pressures. (**a**) Rock content 30%. (**b**) Rock content 40%. (**c**) Rock content 50%.

**Table 1 materials-16-03698-t001:** Main physical property indexes of S-RM.

*R*/%	*w_na_*/%	*G_s_*	*w_op_*/%	*ρ_d max_*/(g·cm^−3^)
47.29	2.57	2.72	7.94	1.91

*R* = rock content; *w_na_* = natural water content; *G*_s_ = specific gravity; *w_op_* = optimum water content; *ρ_d max_* = maximum dry density.

**Table 2 materials-16-03698-t002:** Mineral composition of rock (in %).

Quartz	Illite	Albite	Kaolinite	Chlorite	Calcite	Hematite
48.8	22.0	17.9	2.7	5.5	1.8	1.2

**Table 3 materials-16-03698-t003:** Main physical parameters of S-RM samples.

*R*/%	*n*	*w*/%	*ρ_d max_*/(g·cm^−3^)	*ρ_t_*/(g·cm^−3^)
20	0.353	5	1.71	1.65
30	0.318	5	1.78	1.75
40	0.283	5	1.87	1.85
50	0.249	5	1.98	1.95

*R* = rock content; *n* = porosity; *w* = water content; *ρ_d max_* = maximum dry density; *ρ_t_* = target dry density.

**Table 4 materials-16-03698-t004:** Statistics of the electrical resistivity values.

*R*/%	*ER*/(Ω·m)	Range/(Ω·m)	Mean/(Ω·m)	*SD*/(Ω·m)	*CV*/%
20	47.18, 46.85, 47.25, 47.56	0.71	47.21	0.25	0.53
30	44.07, 44.28, 43.92, 44.45	0.53	44.18	0.20	0.46
40	39.90, 40.47, 40.55, 39.88	0.67	40.20	0.31	0.77
50	38.42, 38.16, 37.87, 38.07	0.55	38.13	0.19	0.50

*R* = rock content; *ER* = electrical resistivity (from four measurements); *SD* = standard deviation; *CV* = coefficient of variation.

## Data Availability

Not applicable.

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
