# Peer review of "Analysis of Electrical Resistivity Characteristics and Damage Evolution of Soil–Rock Mixture under Triaxial Shear"

_materials, 2023, doi:10.3390/ma16103698_

Round 1

Reviewer 1 Report

The paper presents an investigation of the modified triaxial apparatus used to measure the electrical resistivity of the soil-rock mixture subjected to triaxial shear, and the electrical resistivity method was used to evaluate the change of damage variable that quantifies the degree of damage. The document is organized and concise. English is good but grammatical errors, need to be addressed. The topic is relevant to the current research community. Please, see the comments and include modifications to the manuscript accordingly.

1.      The originality of the work is not obvious, please emphasize the problem statement, and explain the originality that is not done before. Please emphasize your contribution

2.      The introduction and literature review are too short. Please add 1-2 pages of recent work. Don’t write [1-2 or 4-5] please explain each reference separately and add the following

a.      Khurshid, I., Afgan, I. 2022. Geochemical Investigation of Electrical Conductivity and Electrical Double Layer based Wettability Alteration during Engineered Water Injection in Carbonates. Journal of Petroleum Science and Engineering. 215,110627.

b.      Yao, Y., Li, J., Ni, J., Liang, C. and Zhang, A., 2022. Effects of gravel content and shape on shear behaviour of soil-rock mixture: Experiment and DEM modelling. Computers and Geotechnics, 141,104476.

c.      Khurshid, I., and Fujii, Y. 2021. Geomechanical analysis of formation deformation and permeability enhancement due to low-temperature CO2 injection in subsurface oil reservoirs. Journal of Petroleum Exploration and Production Technology, 11(4): 1915:1923.

3.      Please read the paper a number of times and fix grammatical errors.

4.      Please write the definition of damage, it is not clear what the author means. Does damage mean compaction? Or decrease in permeability?

5.      Increase the number of figures and please provide a schematic diagram

6.      Please provide references for the data in all equations 1-18 in the paper. Also, provide the source of  Fig. 9.

7.      In the validation section, please also compare your results with the results given in the literature. Validating one experimental data with one developed model doesn’t make sense. Without the developed technique validation with experimental or field data from other sources, the paper is worthless.

8.      Please show the benefits of the suggested report and compare them with the existing available techniques.

9.      Rewrite the conclusion in a descriptive way, it’s too long and emphasize the objectives and benefits of the study.

1.      Please read the paper a number of times and fix grammatical errors.

Reviewer 2 Report

I read your effort and novelty. Before publishing several comments are addressed. 

1.     Abstract is not well organized, so the reviewer recommend that follow that pattern [background] -> [objective] -> [process] -> [results] -> [contribution]

2.     In ‘Introduction’, the author conducted the literature review to prove the excellence of research, However, the author should explain the superiority of research comparing with other researches too.

3.      I recommend the authors should insert such figure which can express the materials characteristics such as SEM and XRM

4.     Distribution of each measured data could not be only indicated by one value. The error bar should be added.

Round 2

Reviewer 1 Report

Please read the paper a couple of times and fix grammar and typos.

Please read the paper a couple of times and fix grammar and typos.